# Assessment of the viability of photovoltaic system implementation on the New Media Tower of Universitas Multimedia Nusantara using PVSyst software: A feasibility study

Fahmy Rinanda Saputri[1]*, Nicholas Robert[1], Agie Maliki Akbar[2]

1 Department of Engineering Physics, Universitas Multimedia Nusantara, Tangerang, Indonesia,
2 Department of Engineering Physics, Surya University, Tangerang, Indonesia

* fahmy.rinanda@umn.ac.id

## Abstract

The urgency of addressing climate change, exacerbated by greenhouse gas emissions, necessitates sustainable solutions, including green building practices and renewable energy adoption. This study focuses on the feasibility of implementing solar photovoltaic systems at Universitas Multimedia Nusantara (UMN), particularly in Building C, known as the New Media Tower, which is designed with green building principles. Solar energy, an increasingly prominent renewable source, presents a viable solution to reduce carbon footprints. Before installation, thorough simulations using software like PVSyst are essential to predict energy output and evaluate system efficiency. Several studies have explored PV system feasibility using simulations, highlighting the importance of software selection for accurate assessments. Building C offers potential locations for PV installation, with the rooftop being a primary consideration due to its expansive area and minimal shading. The rooftop PV system simulation shows an annual energy production of 202 MWh, close to the target of 209.64 MWh, while the parking area system only generates 64.5 MWh/year. Technical evaluations reveal that only the rooftop PV system meets electricity generation targets, highlighting its superiority over the parking area system. Financial analysis demonstrates the rooftop system's viability, with a payback period of 8.2 years and a return on investment (ROI) of 115.8%. Although the upfront investment is substantial, the long-term benefits justify implementation. Overall, this study underscores the technical and financial feasibility of rooftop photovoltaic systems on Building C at UMN, offering valuable insights for sustainable energy initiatives in academic institutions.

## 1 Introduction

Currently, climate change and its negative impacts are among the greatest challenges facing humanity [1,2]. One of the major contributors to climate change is the production of greenhouse gases stemming from human activities [3,4]. One such activity is the emission of carbon dioxide ($CO_2$) [5], also known as carbon footprint [6–9]. One way for humans to reduce their

**Data Availability Statement:** All relevant data are within the paper and its Supporting Information files.

**Funding:** The author(s) received no specific funding for this work.

**Competing interests:** The authors have declared that no competing interests exist.

carbon footprint is by implementing green building practices [10,11]. The concept of a green building is a means to achieve sustainability. Green Building involves the holistic approach to the design, construction, operation, and upkeep of buildings, considering factors like efficient use of natural resources, indoor air quality, and the occupants' well-being [12–14].

In the area of Universitas Multimedia Nusantara (UMN), the New Media Tower, also known as Building C, is one of the buildings designed according to green building principles. Building C is one of the two buildings on the UMN campus that features a double facade, an aluminum perforated shell enveloping the entire exterior of the building, regulating the sunlight entering. This double facade allows Building C to maintain adequate natural lighting while ensuring thermal comfort for occupants without excessive use of active cooling systems. Besides energy savings, another way to reduce a building's carbon footprint is by utilizing renewable energy sources [15–18] such as utilized solar energy [19]. Solar energy generation is on the rise, primarily due to advancements in solar energy technologies, which researchers continuously improve to achieve greater energy conversion efficiency [20]. An example is the installation of solar photovoltaic (PV) panels integrated into or around the building, which can generate some or all of the electricity needed by the building.

Before the actual installation of a solar power generation system, simulation using software is necessary. This is done to understand the system's performance comprehensively, predict the energy output and evaluate factors such as efficiency and optimal placement of solar panels. Thus, simulation enables better planning before actual implementation [21–23].

There are several studies conducting simulations of energy. Studies Kartikasari et al. and Shukla et al. share similar objectives, which are to evaluate the feasibility of solar photovoltaic power generation systems at their respective locations using simulation. However, a study by Kartikasari et al. was conducted in Surabaya, Indonesia, focusing on a 3 kWp rooftop photovoltaic system [24], while a study by Shukla et al. was conducted at MANIT, Bhopal, India, focusing on a 110 kWp rooftop photovoltaic system [25]. Both studies utilized SolarGIS PV Planner for simulation, although Study Kartikasari et al. also used RETScreen.

Then there is another relevant study, Siregar et al.'s research optimizes the design of solar panels around the Artificial Lake at USU using PVSyst, with the optimal design using Si-Mono 310 Wp solar panels, resulting in 144.21 MWh/year of energy [26]. There is a wide variety of software available for simulating the construction of photovoltaic solar power systems, allowing for comparison between different software packages for the same research object. This was done by some researchers [27–29]. They used PVSyst and RETScreen software to estimate the electricity generated. In addition to PVGIS, RETScreen, and PVSyst, there is also PVWatts software that can be used to estimate PV electricity production, as demonstrated by [30].

In addition, these studies discuss the importance of performance and economic analysis in the implementation of photovoltaic systems to support the transition to renewable energy. Research in Islamabad, Pakistan, using PVSyst to design PV systems in luxury villas and high-rise buildings, revealed that payback analysis and performance ratios are critical to assessing financial viability, both through direct purchase and bank financing [31]. In India, research showed that with government support, residential rooftop PV systems become more economically viable, and simulations using PVSyst compared different PV module technologies based on energy performance and long-term economic analysis [32]. Research in Bangladesh emphasized the efficiency and carbon emission reduction potential of grid-connected PV systems, where the systems have the potential for cost savings and generate revenue from selling electricity to the grid [33]. The findings suggest that a comprehensive technical and economic feasibility analysis before PV installation is essential to ensure sustainable energy benefits in various locations. Based on this research, we propose research related to the feasibility study of

the photovoltaic system at the New Media Tower at Universitas Multimedia Nusantara using PVSyst. This analysis will include technical and economic evaluations.

Although Universitas Multimedia Nusantara (UMN) has implemented the green building concept in Building C through the use of double facades that help reduce energy consumption, efforts to achieve more holistic sustainability can still be improved by utilizing renewable energy, especially solar energy. Currently, there is no comprehensive study on the potential of solar energy on UMN's campus, especially regarding the installation of photovoltaic systems in Building C. To support the target of reducing the carbon footprint and increasing energy efficiency in this building, research is needed on the technical and financial feasibility of a solar power generation system. This research is also important to determine the most optimal location around Building C that can produce maximum energy, and evaluate whether the target of providing 20% of the building's electricity needs through solar power can be achieved.

The main focus of this research is on the technical and financial feasibility study of a solar power generation system in Building C or New Media Tower, Universitas Multimedia Nusantara (UMN) using PVSyst. The objective is to evaluate the potential locations around the building for solar panel installation. The target of this research is to fulfill 20% of Building C's electricity demand through solar energy. Several locations around Building C will be analyzed to assess the technical and financial feasibility of installing a solar power generation system, making it possible to achieve the target of providing 20% of the building's electricity needs from solar power. Furthermore, this research will provide an in-depth analysis of the technical and financial feasibility of installing photovoltaic systems at selected locations in and around UMN Building C.

The novelty of this research lies in the integration of renewable energy technology, especially photovoltaic systems or solar power plants, in the concept of green building in tropical areas, with objects in university buildings in Indonesia, especially in UMN. The use of PVSyst simulation software to evaluate the solar energy potential of UMN's Building C is an approach that has not been widely applied in Indonesian educational institutions. This research also offers results that allow implementation to achieve energy reduction targets through technical and economic approaches.

## 2 Methods

This section describes the methodology used for the technical and economic analysis of the solar photovoltaic (PV) system on Building C at Universitas Multimedia Nusantara (UMN). The analysis process was carried out in two major phases: technical and economic evaluation to assess project feasibility. This analysis is conducted by using PVSyst software.

### 2.1 Approach

To conduct research on the feasibility of a photovoltaic system at UMN's Building C or New Media Tower, this study utilizes a methodological framework that integrates technical and economic analysis. Using PVSyst software, simulations were conducted to predict solar energy output, taking into account local solar irradiation levels. The economic analysis included payback periods and life cycle cost assessments based on initial investment and long-term energy savings. The solar irradiation data for the simulation was sourced from Meteonorm, which provides reliable historical climate data specific to the building site. The following are the limitations for the exploration and evaluation of potential photovoltaic solar power systems presented in this research:

- Location: The designated area for assessment is in the vicinity of the New Media Tower (Building C) at Universitas Multimedia Nusantara, situated on Scientia Boulevard Gading, Curug Sangereng, Serpong, Tangerang Regency, Banten Province, Indonesia [34].

- Software: PVSyst software was utilized to conduct simulations of solar irradiation, near-shading, and economic analyses.

- Solar Irradiation Data Source: Solar irradiation data from the Meteonorm database spanning the years 2016–2021 were utilized. This data, synthesized directly through the Meteo Database feature of PVSyst software [35], was tailored to the specific location of Building C at Universitas Multimedia Nusantara.

- Generation Target: The targeted generation for the designed photovoltaic solar power systems is a minimum of 20% of the electricity consumption of Building C.

## 2.2 Data

Initially, data collection involved a site survey of Building C and its surroundings. Data included the location as simulation objects, solar irradiance, and shading analysis. The survey also covered information on existing infrastructure that may impact photovoltaic system installation. This information was critical for system orientation and shading simulations. For solar irradiance data, historical meteorological data for the region was obtained through the Meteonorm database, which provides location-specific irradiance values necessary for accurate simulations. The survey also took into account local weather conditions, which affect PV system performance.

Throughout the New Media Tower of Universitas Multimedia Nusantara, several locations with potential for photovoltaic installation were identified. The following will discuss two selected locations in more detail, considering the advantages and disadvantages of each location. The data presented will serve as the basis for design and near shading using PvSyst.

The rooftop of Building C in Fig 1 is one of the locations worth exploring because it is almost entirely free from shading effects from surrounding buildings, especially from Building C itself. The rooftop area is also relatively large, approximately 1,000 m$^2$, thus having the highest potential to produce electricity in sufficient quantities to meet the target.

However, for PV system installation considerations, there are several challenges that make implementation difficult and raise administrative obstacles for solar panel installation. First, this part of the building's roof is used as the headquarters of Skystar Ventures, designed to incorporate sunlight as the main source of lighting. Installing solar panels would obstruct most of the sunlight falling on the roof floor and increase electricity consumption for lighting on that floor. Furthermore, obtaining operational permits for installation and maintenance will be more difficult compared to other locations because, in general, the rooftop is only accessible to members of Skystar Ventures. Moreover, the unconventional facade shape and high rooftop position also complicate the PV installation and maintenance processes. Although the rooftop location has some drawbacks, its large area and minimal shading potential make it the most suitable for primary evaluation, especially for supplying electricity for the substantial electricity needs of Building C.

The outdoor motorcycle parking area is shown in Fig 2 located next to Buildings C and D. Because it is almost at the same level as the ground floor of the building, installing and maintaining PV systems is easier and more affordable than on the roof. However, the motorcycle parking area has two main obstacles: the presence of many trees and buildings C and D nearby, and the limited area of the motorcycle parking space. This limited area, approximately only

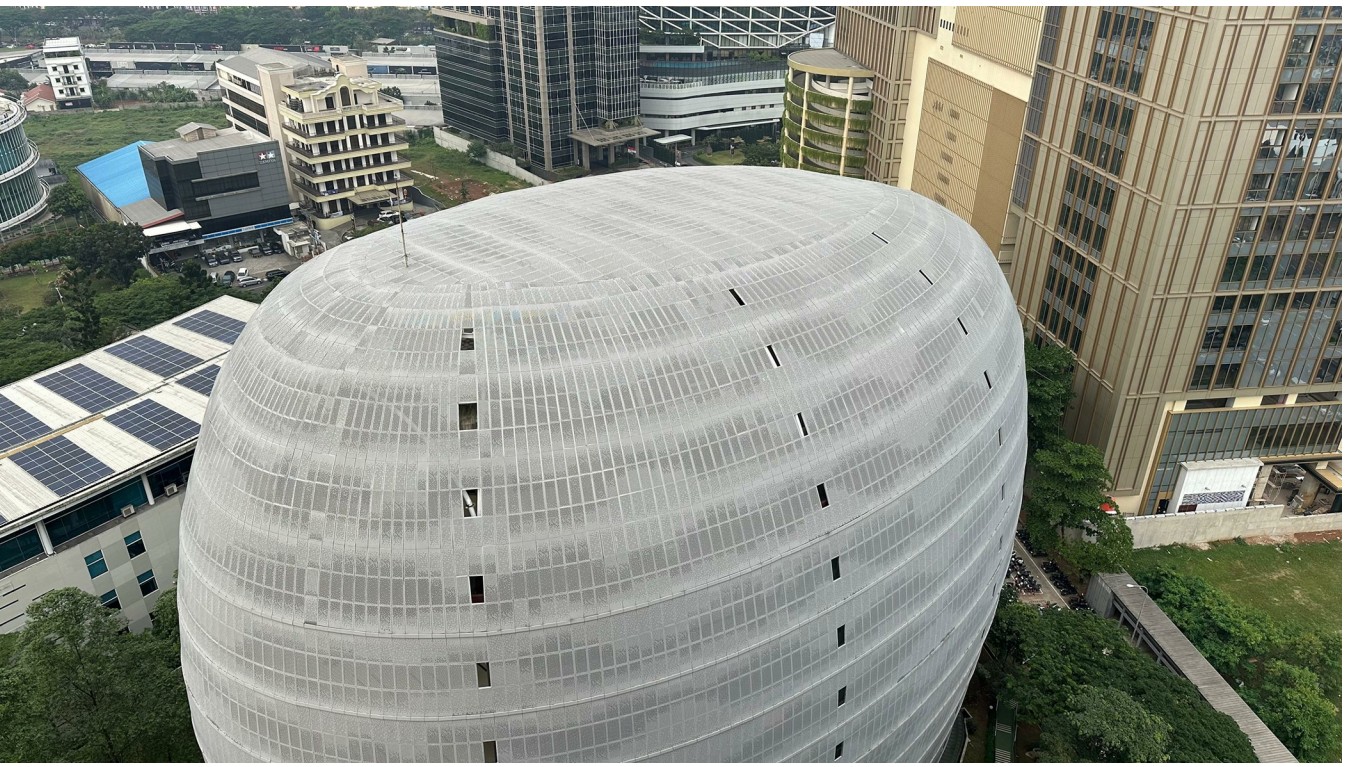

**Fig 1. Rooftop of Building C Universitas Multimedia Nusantara (Picture taken by the author (Fahmy Rinanda Saputri) and can be published under CC BY 4.0 license).**

400 m$^2$, reduces production output, and the presence of obstacles such as trees and buildings that create shading reduces the potential for electricity production here. Nevertheless, this location still appears feasible for analysis because if it can achieve production targets, installing and maintaining solar panels here will be much cheaper.

Electricity consumption data is obtained directly from the UMN Building Management. The raw data obtained is as follows in Table 1.

For analysis and target determination purposes, this data needs to be converted into annual data. Assuming that the electricity consumption in these three months represents a typical three-month period for electricity consumption throughout the year, and all buildings in Universitas Multimedia Nusantara (A, B, C, and D) consume electricity in equal amounts, the target value can be calculated as follows in Table 2. The target electricity generation value is 209,640 kWh per year or 209.64 MWh per year.

## 2.3 System design and simulation using PVSyst

Once the site-specific data was gathered, system design and simulations were carried out using PVSyst software, a widely recognized tool for PV system design and energy yield prediction. The following steps were taken:

- Defining photovoltaic system parameters

- Array sizing

- Shading analysis

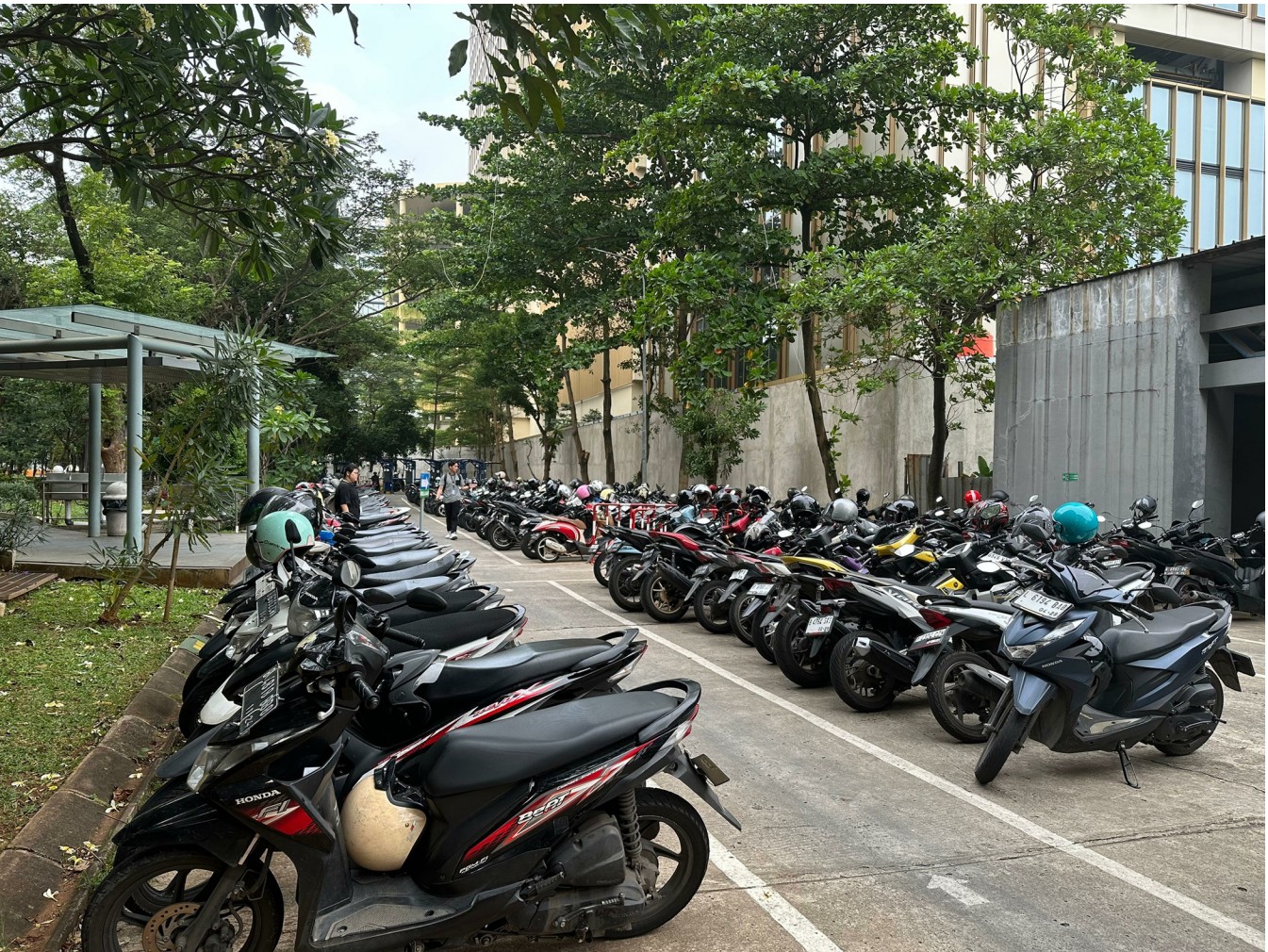

**Fig 2. The outdoor motorcycle parking area (Picture taken by the author (Fahmy Rinanda Saputri) and can be published under CC BY 4.0 license).**

- System orientation
- Energy yield calculation

## 2.4 Economic analysis

To evaluate the financial feasibility of the rooftop PV system, an economic analysis was conducted based on the following steps:

- Cost estimation

**Table 1. UMN electricity consumption January-March 2023 (from UMN building management).**

| Periods | Electricity consumption (kWh) | |
|---|---|---|
| | **Off-peak hours** | **Peak load hours** |
| January 2023 | 303,000 | 30,540 |
| February 2023 | 293,940 | 32,040 |
| March 2023 | 349,560 | 39,120 |

**Table 2. Electricity generation target calculation.**

| Periods | Electricity consumption (kWh) | | |
|---|---|---|---|
| | **Off-peak hours** | **Peak load hours** | **Total** |
| January 2023 | 303,000 | 30,540 | 333,540 |
| February 2023 | 293,940 | 32,040 | 325,980 |
| March 2023 | 349,560 | 39,120 | 388,680 |
| Total from Jan-March 2023 | 946,500 | 101,700 | 1,048.200 |
| **Data processing** | | | |
| Electricity consumption in 2023 | 3,786,000 | 406,800 | 4,192,800 |
| Annual electricity consumption per building | 946,500 | 101,700 | 1,048,200 |
| 20% of the annual electricity consumption per building | 189,300 | 20,340 | 209,640 |

- Operational cost

- Revenue assumptions

- Net Present Value (NPV) calculation

- Payback period and ROI

## 3 Results and analysis

By incorporating the data obtained through site surveys, simulations were conducted using PVSyst software. Here is a detailed breakdown of all the simulation results. Below is a description of the types of equipment used in the simulation using PVSyst software.

- The solar panels used are Canadian Solar (CSI Solar) model CS3K-290P 1500V P4, with a capacity of 290 Wp per panel [36]. These panels are used for both location variations and are available internationally at a price range of US$0.24–0.26 per Wp (depending on the order size).

- The inverter used is the Ginlong Technologies inverter, model Solis-30K, with a capacity of 30 kW [37]. The 30 kW model is used for both location variations. This inverter is available internationally at a unit price of US$1,950.00.

- Wiring is done on a mass scale and only depends on the number of inverters, as solar panels have their connector cables that can be used to arrange the panels in series. It is assumed that the number of cables used is equal to the number of inverters multiplied by the number of strings.

Before conducting simulations, it's necessary to determine the location of the installation site, equipped with irradiance data. The location is determined using the Meteo database in the main project menu. In the Main meteo data section, the Geographical sites button is selected, then click the New button at the bottom. A world map will appear, which can be clicked to select. For simulations on the rooftop of the building is utilized two types of orientations, there are Orientation 1: tilt 15˚, azimuth 10˚ and Orientation 2: tilt 30˚, azimuth 10˚. For simulations in the parking area, where there are already roofs suitable for installation, only one orientation is used, with a tilt of 15˚ and azimuth of 70˚. After determining the optimal orientation for PV panels for a particular variant, the next step is to determine the type and quantity of solar panels, inverters, and the arrangement of series and parallel connections.

In this study, the Canadian Solar CS3K-290P 290 Watt poly Solar Panel Module was used. This panel is known as a polycrystalline panel with a capacity of 290 Watts which is used to

maximize the absorption of solar energy. Meanwhile, the inverter used is a Ginlong technologies 30kW 200-800V TL 50/60 Hz Solis-30K inverter, which has an output voltage of 200-800V. This inverter functions to convert direct current (DC) generated by solar panels into alternating current (AC) which can be used by the building's electrical network.

To design large-scale photovoltaic solar power systems, particularly, the most important parameter to know and input is the area of land available for module installation. Another way to achieve the same goal is by using the planned power capacity in the Wp (Watt-peak) of photovoltaic modules. After determining one of the two design options (land area or system power plan), the array sizing process can begin, followed by the creation of a single-line diagram. In this case, the land area used is 800 m$^2$ for building C's roof and 300 m$^2$ for motorcycle parking, so that when the project is implemented, modules experiencing excessive shading can be partially relocated to minimize shading.

Array sizing is the process of determining the number of solar panels based on the selected panel model, inverter, and either land area or system power plan to meet the specified target. This process can be done manually by adjusting the numbers in the Number of modules and strings section, or automatically by clicking the Resize button in the Pre-Sizing Help section. For this system, the following array sizing limits are used:

- Rooftop: 20 modules in series, and 24 module strings to cover an area of 798 m$^2$, with 16 modules oriented as Orientation 1 (tilt 15˚, azimuth 10˚) and 8 modules as Orientation 2 (tilt 30˚, azimuth 10˚). Four inverters of 30kW are used.

- Parking: 20 modules in series, and 9 module strings to cover an area of 299 m$^2$. Two inverters of 30kW are used (slightly oversized).

The system design process continues with the creation of a single-line diagram (SLD). SLD is a simple technical drawing representing the PV system design. In PVSyst, SLDs are automatically generated and accessible through the Single-line diagram button in the system definition menu.

Shading diminishes the electrical output of a photovoltaic (PV) system [38–40]. To estimate shading effects, the near shading feature in the PVSyst software is utilized [41], allowing users to create a simple three-dimensional (3D) model representing buildings, trees, and other obstacles that may reduce sunlight exposure on panels. Here is the near-shading menu for the roof variant.

To determine the amount of shading and when shading occurs for each panel installation, a near-shading scheme is created for both variants. Fig 3 below shows the 3D results used as input for the roof near shading simulation. Meanwhile, Fig 4 below shows the 3D results used as input for the outdoor motorcycle parking near-shading simulation. It can be observed that the number of objects in this near shading scheme is much higher because the placement of solar panels low to the ground and close to the surface increases the potential for near shading.

After designing the system following these steps, performance simulation and system production results can be initiated. The simulation results depend on the information from each previous step of system orientation, design, and near-shading schemes. The estimated system production is 2MWh/year for the rooftop and 64.5 MWh/year for parking area.

## 3.1 Feasibility study

Among the two variants of solar photovoltaic system (PV) installation cases discussed, only the rooftop variant can approach the production target of 209.64 MWh/year. The simulation results indicate that the designed rooftop PV system can generate 202 MWh/year, which is very close to the target. On the other hand, the designed PV system in the parking area can

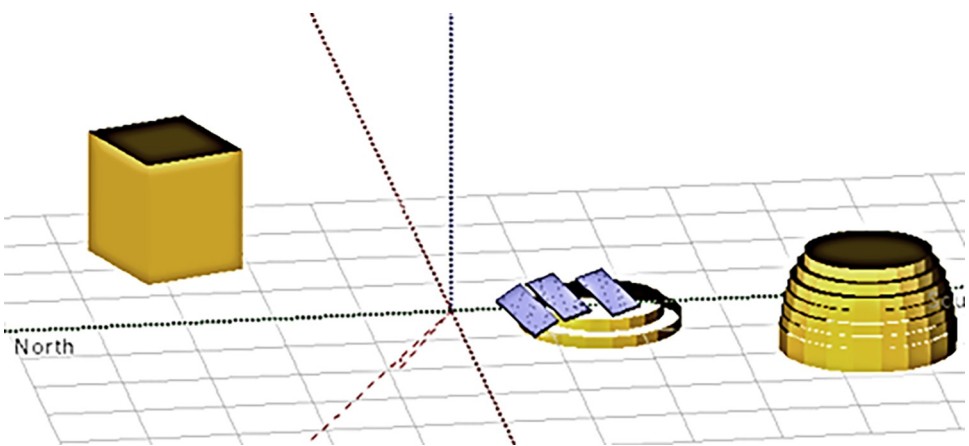

**Fig 3. Scheme of near shading roof variants.**

only generate 64.5 MWh/year, significantly smaller than the target and the simulation results of the rooftop PV system.

Furthermore, looking at production performance, the parking system only produces 3.38 kWh/kWp/day, which means the production efficiency of the parking PV system is reduced by about 15% compared to the rooftop PV system, which produces 3.98 kWh/kWp/day. In short, the location of the PV system in the parking area is quite poor, especially due to the

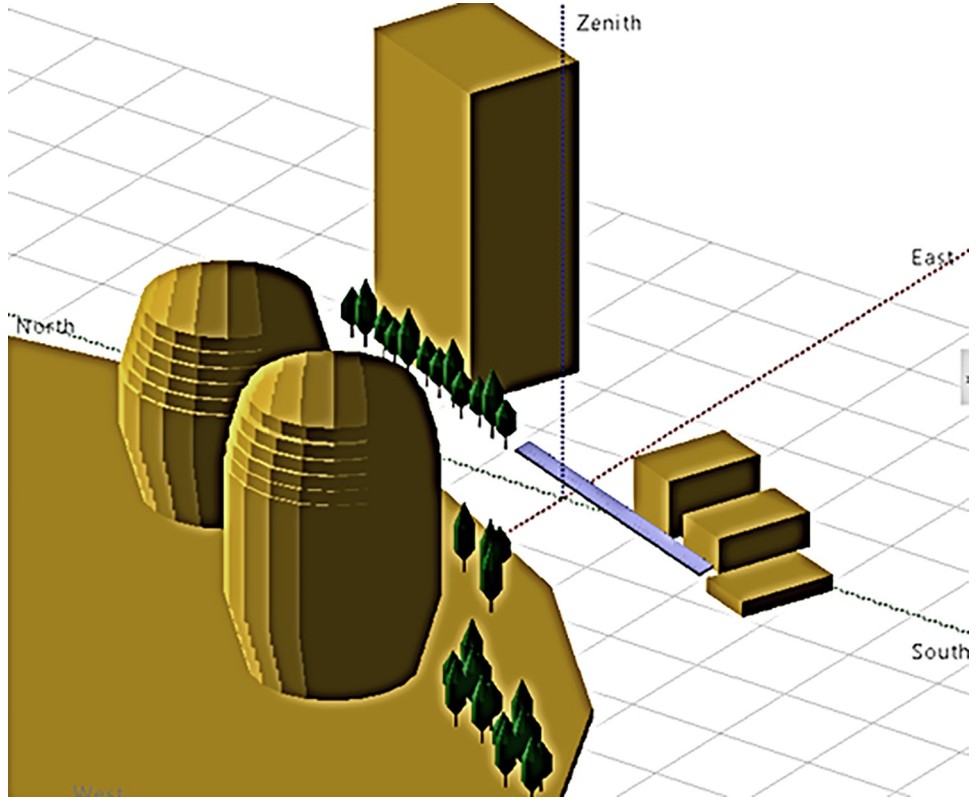

**Fig 4. Scheme of near shading parking variants.**

**Table 3. Division of project installation costs.**

| Cost | Amount | Price per item (Rp) | Total price (Rp) |
|---|---|---|---|
| **PV modul** | | | |
| CS3K-290P1500VP4 | 480 | 1,007,869.52 | 483,777,371.51 |
| Modul buffer | 480 | 290,000.00 | 139,200,000.00 |
| **Inverter** | | | |
| Solis-30K | 4 | 27,108,214.78 | 108,432,859.13 |
| **Other conponents** | | | |
| Screws and other accessories | 480 | 100,000.00 | 48,000,000.00 |
| Cables | 96 | 300,000.00 | 28,800,000.00 |
| Combiner box | 4 | 2.000,000.00 | 8,000,000.00 |
| **Administration** | | | |
| Permits and other administration | 1 | 10,000,000.00 | 10,000,000.00 |
| **Installation** | | | |
| Per modul | 480 | 100,000.00 | 48,000,000.00 |
| Per inverter | 4 | 4,000,000.00 | 16,000,000.00 |
| **Tax** | | | |
| Value Added Tax | 11% | 779,410,230.64 | 85,735,125.37 |
| **Total** | | | **975,945,356.01** |

presence of many objects such as buildings and trees that contribute to shading around this parking area. Therefore, among the two locations, the rooftop of Building C itself stands out as the suitable site for PV system installation.

## 3.2 Economics analysis

To analyze whether the rooftop PV system is financially viable. An economic analysis could be conducted with the assistance of PVSyst software [42,43]. Table 3 below shows the breakdown of project installation costs assumed. Then, the operating costs of the system assumed are outlined in Table 4 below. To calculate the financial feasibility of the project, the Net Present Value (NPV) can be calculated. It is assumed that the project begins in 2024, with a duration of 20 years, inflation rate of 3%, and a discount rate of 0%. It is also assumed that the investment is fully funded by UMN itself, so there is no need to calculate loan interest.

As revenue, feed-in tariffs are used as a representation of the amount of electricity saved by UMN. Electricity saved during off-peak hours (LWBP) is valued at Rp1,035.78/kWh, while electricity saved during peak load hours (WBP) is valued at Rp1,553.67/kWh. However, since peak load hours in Indonesia occur at night (18:00–22:00) and the designed PV system does

**Table 4. Division of project operational costs.**

| Cost | Annual cost (Rp/year) |
|---|---|
| **Maintenance** | |
| Inverter depreciation | 5,000,000.00 |
| Salary | 30,000,000.00 |
| Service | 20,000,000.00 |
| Cleaning | 10,000,000.00 |
| **Administration** | |
| Administration and accounting | 1,000,000.00 |
| **Total** | **66,000,000.00** |

not have an energy storage system, all electricity generation is counted during off-peak hours when the sun is shining.

The economic analysis of the photovoltaic (PV) system installation at Universitas Multimedia Nusantara (UMN) was conducted by breaking down the total installation costs into various components. The total costs included expenses for solar panels, inverters, wiring, installation, and administrative fees. For instance, 480 units of Canadian Solar CS3K-290P panels were priced at Rp1,007,869.52 each, amounting to Rp483,777,371.51. Additionally, 4 units of Ginlong Technologies Solis-30K inverters were included at a total cost of Rp108,432,859.13. Other components such as screws, cables, and combiner boxes contributed to a total of Rp393,736,479.64, along with administrative costs estimated at Rp10,000,000 for permits and other administrative tasks. A value-added tax (VAT) of 11% was also included, resulting in Rp85,735,125.37.

Annual operational costs were estimated to total Rp66,000,000, covering expenses such as inverter depreciation, maintenance, salaries, and cleaning services. Revenue generation was assessed based on savings in electricity costs due to the self-consumption of generated energy. Feed-in tariffs were calculated using rates for off-peak hours (LWBP) at Rp1,035.78/kWh and peak load hours (WBP) at Rp1,553.67/kWh, although only off-peak rates applied since generation occurs during these hours.

With these assumptions, costs, and revenues, the financial results are obtained. In the financial results, the project will pay back its entire investment within 8.2 years; in short, this project has a payback period of 8.2 years. Additionally, after 20 years, the project will bring in revenues of around Rp1.1 billion, or equivalent to a return on investment (ROI) of 115.8%. Thus, the rooftop PV system project in Building C is technically and financially feasible.

To evaluate financial feasibility, the Net Present Value (NPV) was calculated, assuming a project duration of 20 years, an inflation rate of 3%, and a discount rate of 0%. It was assumed that UMN would fully fund the investment, thus eliminating the need for loan interest calculations. Financial indicators were also assessed, revealing a payback period of approximately 8.2 years for the project, with projected revenues of around Rp1.1 billion after 20 years, leading to a return on investment (ROI) of 115.8%.

Based on the result and discussion in this study, the rooftop PV system in UMN produced about 202 MWh/year, close to the target of 209.64 MWh/year, while other studies in locations with higher solar radiation reported production above 250 MWh/year. The efficiency of the system in the parking lot is only 3.38 kWh/kWp/day, 15% lower than the rooftop (3.98 kWh/kWp/day). Other studies have shown a 10–20% drop in efficiency in shaded locations. The study showed a payback period of 8.2 years and an ROI of 115.8% in 20 years, which is in line with other studies reporting payback periods between 5 to 10 years.

## 4 Conclusions

From field surveys, two locations were evaluated using PVSyst simulations: the rooftop of Building C and the motorbike parking area. After technical analysis, it was found that only the rooftop PV system design on Building C could approach the target of 20% of Building C's electricity consumption, with a value of 202 MWh/year compared to the target of 209.64 MWh/year. The parking area PV system design only produced 64.5 MWh/year due to limited space and numerous objects causing near shading.

After evaluating the technical feasibility, the PV system project designs were also assessed financially. With some assumptions about installation and operating costs, it was found that the payback period is 8.2 years. Assuming the PV system operates for 20 years after installation (without changes in weather conditions, electricity costs, or production quantity), the project's

NPV value is Rp1.1 billion, resulting in an ROI of 115.8%, indicating that the project is worth implementing. However, the relatively long payback period makes the required investment feel heavy, amounting to nearly Rp975 million. In short, this project seems feasible for implementation at the related location.

## Acknowledgments

We would like to express our deepest gratitude for the opportunity to conduct research and publication. We highly appreciate the support and collaboration provided throughout this process. Hopefully, the results of this research will be beneficial for the development of science and progress for society. Thank you for all the opportunities and support provided by Universitas Multimedia Nusantara.

## Author Contributions

**Conceptualization:** Agie Maliki Akbar.

**Data curation:** Nicholas Robert.

**Formal analysis:** Fahmy Rinanda Saputri.

**Methodology:** Nicholas Robert.

**Software:** Nicholas Robert.

**Supervision:** Agie Maliki Akbar.

**Validation:** Fahmy Rinanda Saputri.

**Visualization:** Nicholas Robert.

**Writing – original draft:** Nicholas Robert.

**Writing – review & editing:** Fahmy Rinanda Saputri.

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
