## [Decision Letter · Decision Letter 0]

29 Sep 2024

PONE-D-24-29663Assessment of the viability of photovoltaic system implementation on the New Media Tower of Universitas Multimedia Nusantara using PVSyst Software: A feasibility studyPLOS ONE

Dear Dr. Saputri,

Thank you for submitting your manuscript to PLOS ONE. After careful consideration, we feel that it has merit but does not fully meet PLOS ONE’s publication criteria as it currently stands. Therefore, we invite you to submit a revised version of the manuscript that addresses the points raised during the review process. Please submit your revised manuscript by Nov 13 2024 11:59PM. If you will need more time than this to complete your revisions, please reply to this message or contact the journal office at plosone@plos.org. Please include the following items when submitting your revised manuscript:A rebuttal letter that responds to each point raised by the academic editor and reviewer(s). You should upload this letter as a separate file labeled 'Response to Reviewers'.A marked-up copy of your manuscript that highlights changes made to the original version. You should upload this as a separate file labeled 'Revised Manuscript with Track Changes'.An unmarked version of your revised paper without tracked changes. You should upload this as a separate file labeled 'Manuscript'.

We look forward to receiving your revised manuscript.

Kind regards,

Morteza Taki, Ph.D

Academic Editor

PLOS ONE

Journal Requirements:

5. We note that Figure 1 in your submission contain copyrighted images. All PLOS content is published under the Creative Commons Attribution License (CC BY 4.0), which means that the manuscript, images, and Supporting Information files will be freely available online, and any third party is permitted to access, download, copy, distribute, and use these materials in any way, even commercially, with proper attribution. For more information, see our copyright guidelines: http://journals.plos.org/plosone/s/licenses-and-copyright.

Reviewers' comments:

Reviewer's Responses to Questions

**Comments to the Author**

1. Is the manuscript technically sound, and do the data support the conclusions?

Reviewer #1: Partly

Reviewer #2: Partly

2. Has the statistical analysis been performed appropriately and rigorously? 

Reviewer #1: Yes

Reviewer #2: N/A

3. Have the authors made all data underlying the findings in their manuscript fully available?

Reviewer #1: Yes

Reviewer #2: No

4. Is the manuscript presented in an intelligible fashion and written in standard English?

Reviewer #1: No

Reviewer #2: No

5. Review Comments to the Author

Reviewer #1: Dear editor, Dr. Taki

I evaluated the paper and find that it needs some revision. Also some specific question are at the below and the authors should answer them.

Reedit the paper based on English language

Vast the introduction section and use some new and related references

Compare the results with other similar researches

Questions

1. What is the primary objective of this study?

2. Why is addressing climate change urgent according to the paper?

3. What are green building practices, and how do they contribute to sustainability?

4. Why was Building C at Universities Multimedia Nusantara chosen for this study?

5. What are the main features of Building C that align with green building principles?

6. How does the double facade of Building C contribute to energy efficiency?

7. What role does solar energy play in reducing carbon footprints?

8. Why are simulations using software like PVSyst essential before actual PV system installation?

9. What are the potential locations for PV installation on Building C?

10. Why is the rooftop of Building C considered the most suitable site for PV installation?

11. What are the challenges associated with installing PV systems on the rooftop of Building C?

12. How does shading affect the performance of PV systems?

13. What are the advantages and disadvantages of installing PV systems in the outdoor motorcycle parking area?

14. What types of solar panels and inverters were used in the simulations?

15. How was the optimal orientation for PV panels determined in the study?

16. What are the key findings from the performance simulations of the rooftop and parking area PV systems?

17. How does the financial analysis demonstrate the viability of the rooftop PV system?

18. What is the payback period for the rooftop PV system, and why is it significant?

19. How does the return on investment (ROI) for the rooftop PV system compare to the initial investment?

20. What assumptions were made in the economic analysis of the PV system project?

21. How does the study address the administrative challenges of installing PV systems on the rooftop?

22. What are the environmental benefits of implementing the rooftop PV system at UMN?

23. How does the study contribute to sustainable energy initiatives in academic institutions?

24. What are the limitations of the study, and how might they impact the results?

25. What future research directions does the paper suggest?

Reviewer #2: PLOS ONE

Assessment of the viability of photovoltaic system implementation on the New Media Tower of Universitas Multimedia Nusantara using PVSyst Software: A feasibility study (Manuscript Number: PONE-D-24-29663).

General comments

It is suggested that the text be revised by a native in English language.

Abstract

Please provide more details (numerical results) in the abstract section.

Keywords

Avoid abbreviations in this section.

Introduction

I believe that the introduction needs to be changed. Introduction should be rewritten focusing on the aim of your work. The statement of the issue and research background is not well done.

- More relevant articles with their most important results should be mentioned.

- The novelty of the work must be written on the more clear and more concise way at the end of introduction section.

Methods

- Methods is incompletely written and needs to be completed according to the relevant sources.

- Calculations should be mentioned in detail. Only the software is mentioned and there is no method.

- Details of economics analysis should be mentioned

Results and discussion

- The results is generally poorly explored. This section should be redesigned in order to explore and discuss the main findings with other international similar works. Also the discussion should be improved.

6. PLOS authors have the option to publish the peer review history of their article (what does this mean?). If published, this will include your full peer review and any attached files.

Reviewer #1: No

Reviewer #2: No

---

## [Author Response · Author response to Decision Letter 0]

15 Nov 2024

Dear Mr. Morteza Taki, Ph.D,

Thank you for your feedback on our manuscript. We have carefully revised the document in accordance with the reviewers' comments. The revised manuscript, response to reviewers, and a marked-up copy with track changes have been uploaded. We appreciate your consideration of the updated submission.

Best regards,

Fahmy Rinanda Saputri

---

## [Decision Letter · Decision Letter 1]

19 Nov 2024

Assessment of the viability of photovoltaic system implementation on the New Media Tower of Universitas Multimedia Nusantara using PVSyst Software: A feasibility study

PONE-D-24-29663R1

Dear Dr. Saputri,

We’re pleased to inform you that your manuscript has been judged scientifically suitable for publication and will be formally accepted for publication once it meets all outstanding technical requirements.

Kind regards,

Morteza Taki, Ph.D

Academic Editor

PLOS ONE

Reviewers' comments:

Reviewer's Responses to Questions

**Comments to the Author**

1. If the authors have adequately addressed your comments raised in a previous round of review and you feel that this manuscript is now acceptable for publication, you may indicate that here to bypass the “Comments to the Author” section, enter your conflict of interest statement in the “Confidential to Editor” section, and submit your "Accept" recommendation.

Reviewer #1: All comments have been addressed

Reviewer #2: (No Response)

2. Is the manuscript technically sound, and do the data support the conclusions?

Reviewer #1: Yes

Reviewer #2: (No Response)

3. Has the statistical analysis been performed appropriately and rigorously? 

Reviewer #1: Yes

Reviewer #2: (No Response)

4. Have the authors made all data underlying the findings in their manuscript fully available?

Reviewer #1: Yes

Reviewer #2: (No Response)

5. Is the manuscript presented in an intelligible fashion and written in standard English?

Reviewer #1: Yes

Reviewer #2: (No Response)

6. Review Comments to the Author

Reviewer #1: I evaluated the paper and I founded that all my comments were addressed in the revised form, so the paper can be accepted.

Reviewer #2: (No Response)

7. PLOS authors have the option to publish the peer review history of their article (what does this mean?). If published, this will include your full peer review and any attached files.

Reviewer #1: No

Reviewer #2: No

---

## [Editor Report · Acceptance letter]

27 Nov 2024

PONE-D-24-29663R1 

PLOS ONE

Dear Dr. Saputri, 

I'm pleased to inform you that your manuscript has been deemed suitable for publication in PLOS ONE. Congratulations! Your manuscript is now being handed over to our production team.

Kind regards, 

on behalf of

Dr. Morteza Taki 

Academic Editor

PLOS ONE